# Relationship Between Morning Blood Pressure Surges and Peripheral Inflammatory Biomarkers in Parkinson’s Disease

**DOI:** 10.3390/biomedicines13020363

**Published:** 2025-02-05

**Authors:** Ummu S. Sari, Seda E. Yildirim, Gulseren Buyukserbetci, Tarik Yildirim, Mesut Sackes, Figen Esmeli

**Affiliations:** 1Neurology Department, Faculty of Medicine, Balikesir University, 10145 Balikesir, Turkey; userpilsari@gmail.com (U.S.S.); fesmeli@hotmail.com (F.E.); 2Cardiology Department, Faculty of Medicine, Balikesir University, 10145 Balikesir, Turkey; sedaelcimdurusoy@gmail.com (S.E.Y.); kdrtarik@gmail.com (T.Y.); 3Necatibey School of Education, Balikesir University, 10145 Balikesir, Turkey; msackes@balikesir.edu.tr

**Keywords:** Parkinson’s disease, morning surge, fibrinogen-to-albumin ratio, CRP

## Abstract

**Background:** Parkinson’s disease (PD) is the second-most prevalent neurodegenerative disorder, often resulting in blood pressure abnormalities due to autonomic dysfunction. The early morning rise in blood pressure, referred to as the morning surge, has been associated with various cardiovascular diseases when exaggerated. This study aims to investigate the relationship between morning blood pressure surge (MBPS) and inflammatory markers in patients with PD. **Methods:** In this retrospective study, we employed 24 h ambulatory blood pressure monitoring alongside the fibrinogen-to-albumin ratio and high-sensitivity C-reactive protein (hs-CRP) as inflammatory markers. The study included fifty idiopathic PD patients and fifty age- and sex-matched control subjects. MBPS was defined as the difference between morning blood pressure (measured two hours after awakening) and the lowest recorded nighttime blood pressure. Body mass index (BMI) was considered as an independent variable. **Results:** Our study found that morning blood pressure surge (MBPS) levels were significantly higher in Parkinson’s disease (PD) patients compared to the control group, suggesting possible autonomic involvement. **Conclusions:** MBPS may indicate autonomic involvement, potentially contributing to cardiovascular and cerebral morbidity and mortality in PD patients. Longitudinal studies with larger sample sizes are warranted to further elucidate this relationship.

## 1. Introduction

Parkinson’s disease (PD) is a complex, progressive neurodegenerative disorder first described by James Parkinson [1]. It is the second-most common degenerative disease, characterized by both motor and non-motor symptoms [2]. Epidemiological estimates based on healthcare utilization data suggest that the incidence of PD ranges from 5 to over 35 new cases per 100,000 individuals annually [3]. The incidence increases five to ten-fold from the sixth to the ninth decades of life, with prevalence also rising with advancing age. Pathologically, PD is characterized by the degeneration of dopaminergic neurons in the substantia nigra pars compacta (SN) of the midbrain, accompanied by the presence of Lewy bodies—cytoplasmic inclusions containing insoluble alpha-synuclein aggregates. However, PD pathology extends beyond the SN to other brain regions and involves non-dopaminergic neurons as well. Historically, it has been estimated that 50–70% of SN dopaminergic neurons have degenerated by the time clinical motor symptoms become detectable [4]. Both hereditary and environmental factors are implicated in its pathogenesis. Notable pathophysiological processes include α-synuclein misfolding and aggregation, mitochondrial dysfunction, reduced protein clearance, neuroinflammation, and oxidative stress. Additionally, concerning the gut–brain axis, the vagus nerve has been proposed to act as a “highway” for transporting aggregated α-synuclein from the gastrointestinal system to the lower brainstem [5].

While the molecular basis of inflammation in PD remains to be fully elucidated, there is increasing evidence supporting its role in disease progression. Abnormalities in both innate and adaptive immune responses—such as elevated levels of pro-inflammatory cytokines and altered immune cell populations (e.g., monocytes and progenitors)—have been documented in PD patients [6,7].

Major motor symptoms include bradykinesia, resting tremor, rigidity, and postural instability. Non-motor symptoms such as cognitive dysfunction, depression, psychosis, apathy, impulse control disorders, and autonomic dysfunction can be equally debilitating. Common autonomic disorders include drooling, orthostatic hypotension (OH), supine hypertension (SH), urinary retention/incontinence, erectile dysfunction, gastrointestinal motility disorders, and hyperhidrosis [8]. The Unified Parkinson’s Disease Rating Scale (UPDRS) is utilized to measure and monitor symptom severity [9]. Although symptomatic treatments can alleviate certain symptoms, they may also lead to undesired side effects such as fluctuations in blood pressure and heart rate. The autonomic nervous system regulates homeostasis by controlling unconscious and involuntary vital functions such as heartbeat, blood pressure, respiratory functions, gastrointestinal motility, urinary functions, and secretory activities [10,11]. Despite the longstanding recognition of autonomic involvement in PD, epidemiological data remain limited due to insufficient attention to symptoms and a lack of clear definitions and objective measurement scales [12]. Symptoms of dysautonomia may manifest as cardiovascular dysfunctions (e.g., OH), urinary issues, sexual dysfunctions, or thermoregulatory abnormalities. OH is the most frequently encountered cardiovascular symptom in approximately 30% of PD patients from the onset of their condition and negatively impacts daily activities [13]. SH occurs in up to 34% of PD patients and is considered indicative of autonomic cardiovascular dysfunction accompanying OH. Long-term implications of SH may include increased risks of myocardial infarction, cognitive dysfunction, and stroke [14,15].

Research on nighttime blood pressure variations in PD has yielded mixed results. The reverse dipping phenomenon indicates nighttime blood pressure abnormalities that serve as biomarkers for cardiovascular autonomic dysfunction. Various tests—including Valsalva maneuver, hyperventilation tests, isometric contraction tests, cold pressure tests, orthostatic tests, head tilt tests, ambulatory blood pressure monitoring (ABPM), and Holter monitoring—are employed to assess autonomic cardiovascular involvement [13]. Blood pressure (BP) exhibits a circadian rhythm characterized by a decrease during nighttime sleep and an increase upon awakening due to sympathetic activation along with renin and cortisol hormone release. This morning the increase in blood pressure is termed morning surge (MS). While MS is a physiological phenomenon, exaggerated MS can become pathological and is associated with myocardial infarction, stroke, and mortality [16]. Factors such as advanced age, abnormal glucose metabolism, smoking habits, endothelial dysfunction, and atherosclerosis may contribute to exaggerated morning blood pressure surges (MBPS) [17]. ABPM over 24 h can document these abnormal activities.

Fibrinogen plays a regulatory role in inflammation and atherosclerosis; high fibrinogen levels are associated with increased inflammation affecting blood viscosity, platelet aggregation, and endothelial function. Conversely, albumin possesses anti-inflammatory and antioxidant properties; low serum levels indicate a heightened systemic inflammatory state. Low albumin levels are independent predictors of elevated blood pressure [18]. The fibrinogen-to-albumin ratio (FAR) has been implicated in the pathophysiological processes underlying cerebrovascular and cardiovascular diseases [19].

C-reactive protein (CRP) is synthesized in the liver and plays a crucial physiological role by binding to phosphocholine on the surfaces of necrotic or apoptotic cells. This interaction activates the immune response, promoting enhanced phagocytic activity by macrophages. In the aftermath of an injury or inflammatory response, CRP levels begin to rise within approximately two hours, with a half-life of about 18 h. Due to its rapid response, CRP is integral to the early stages of the inflammatory response, which is why it is frequently classified as an “acute-phase protein” [20,21]. C-reactive protein (CRP) is one of the most studied biomarkers for inflammation. Its widespread use in clinical practice is attributed to its ease of detection, common application, and good standardization. CRP can detect inflammation independently of its origin and specific pathways, offering advantages in diseases such as Parkinson’s disease (PD) [22].

Parkinson’s disease (PD) is characterized by an increasing incidence with advancing age, typically manifesting after the fifth decade of life. Given this context, the emergence of comorbid systemic diseases known to elevate the risk of atherosclerosis, such as hypertension, diabetes mellitus, and coronary artery disease, in PD patients is not surprising. These comorbidities are associated with increased morbidity and mortality linked to atherosclerosis. The presence of such conditions in Parkinson’s patients may further exacerbate mortality rates. Therefore, we aimed to investigate the morning surge phenomena in this population. Additionally, we sought to elucidate the relationship between morning surge and inflammatory markers such as C-reactive protein (CRP) and fibrinogen-to-albumin ratio (FAR), which are associated with heightened atherosclerosis risk.

Based on the data presented, our objective was to investigate the association between morning blood pressure surge (MBPS) and inflammatory markers in patients with Parkinson’s disease (PD) who have controlled blood pressure and underwent 24-h ambulatory blood pressure monitoring. This inquiry is motivated by findings that indicate an elevated morning surge beyond certain thresholds correlates with an increased risk of mortality. We aimed to explore the implications of morning blood pressure surges that may contribute to heightened morbidity and mortality risks in individuals with PD. Additionally, we examined the correlation between morning surge and inflammatory markers in this population.

## 2. Materials and Methods

This retrospective study was conducted in the Neurology and Cardiology Departments of Balikesir University Faculty of Medicine Hospital. Ethical approval was granted by the Ethics Committee of Balikesir University Faculty of Medicine. For the patient group, we screened 320 Parkinson’s disease (PD) (320 patients out of 753 patient visits) patients who visited the movement disorders outpatient clinic of the neurology department between August 2021 and August 2022 (Figure 1). Fifty PD patients aged over 18 years with 24 h ambulatory blood pressure monitoring (ABPM) recordings (Schiller MT-300 BP, Baar, Switzerland) due to blood pressure fluctuations, along with concurrent inflammatory biomarker data (fibrinogen, albumin, and high-sensitivity C-reactive protein [hs-CRP]), were included in the study. Serum levels of fibrinogen, albumin, and hs-CRP were obtained from the hospital’s digital archive. The fibrinogen-to-albumin ratio (FAR) was calculated by dividing the fibrinogen value by the albumin value. The control group consisted of fifty individuals matched for age and sex, who did not have any systemic diseases and exhibited normal 24-h ambulatory blood pressure monitoring (ABPM) records. These records were obtained from the archive of the Cardiology Department. In patients with PD, Hoehn–Yahr stages and UPDRS motor scores were calculated by a physician specializing in movement disorders immediately prior to the 24 h ambulatory blood pressure measurement, using data from follow-up files. Ambulatory blood pressure monitoring (ABPM) was conducted using the Schiller MT-300 BP device (Baar, Switzerland). Blood pressure measurements were automatically recorded every 15 min during the day and every 30 min at night over a 24 h period, utilizing a cuff attached to the left arm. The sleeping and waking times of all participants were meticulously documented. The system automatically assessed the volunteers’ sleep and wake cycles continuously throughout the monitoring period.

### 2.1. Ambulatory Blood Pressure Measurement

The mean systolic blood pressure (SBP) during the first two hours after awakening was designated as the morning blood pressure surge (MBPS). MBPS was calculated as the difference between the mean SBP over the two hours following awakening and the average of three BP values centered on the lowest nocturnal BP [23]. A review of the literature regarding MBPS revealed that, due to the lack of a precisely defined threshold value, low-value MBPS was categorized as <37 mmHg and high-value MBPS as ≥37 mmHg, based on a meta-analysis involving eight different populations conducted by Li et al. [24].

Patients with kidney or liver diseases or those with inflammatory disorders were excluded from the study to ensure a homogenous patient group.

### 2.2. Statistical Analysis

Data were analyzed using IBM SPSS Statistics for Windows, Version 23.0 (IBM Corp., Armonk, NY, USA). Compliance with normal distribution was assessed using the Kolmogorov–Smirnov test. For normally distributed data, independent two-sample t-tests were employed to compare means, while the Mann–Whitney U test was utilized for non-normally distributed data. Categorical variables were compared between groups using Yates’ correction. To examine the effect of independent variables on morning blood pressure surge (MBPS) values, robust regression analysis was conducted using the MASS package in R, kk 4.4.2. Receiver Operating Characteristic (ROC) analysis was performed to determine the optimal cutoff value for the morning surge parameter. Results are presented as mean ± standard deviation for normally distributed quantitative data and as median (minimum–maximum) for non-normally distributed data. Categorical data are expressed as frequency (percentage). A significance level of *p* < 0.050 was adopted for all statistical tests.

## 3. Results

### Patient Characteristics

Fifty idiopathic Parkinson’s disease (PD) patients and fifty control cases were included in the study. There were no significant differences in age and sex between the PD and control groups (see Table 1 and Table 2). Among the PD patients, eleven individuals (22%) had additional comorbidities: five had diabetes mellitus, three had both diabetes and hypertension, one patient had diabetes, coronary artery disease, and hyperlipidemia, while two had depressive disorders. Demographic data of the participants are summarized in Table 1. Thirty-nine (78%) of PD patients had accompanying illness. Twenty-two (44%) of the group had Hoehn-Yahr stage 1 disease. Patients used oral levodopa, dopamine agonists and jejuna dopa infusion (Table 1).

Thirty-nine (78%) of the PD patients had accompanying illness. Twenty-two (44%) of the group had Hoehn–Yahr stage 1 disease. Patients used oral levodopa, dopamine agonists, and jejuna dopa infusion (Table 1).

UPDRS motor scale median value was 10.00; creatinine median value was 0.92 in PD; 0.70 in control group. Even when both values were in the limits, the difference between the groups was statistically significant (*p* < 0.001) (PD 21.63 control 18.63).

The difference in median values of MBPS between the groups was significant (*p* < 0.005). There was no significant difference between the groups according to age BMI, FAR, diastolic BP, night systolic BP, dipper and night diastolic BP, and average BP (*p* > 0.050). There is also no difference between the groups concerning other parameters (*p* > 0.050).

In the PD group, the effects of independent variables over morning surge are analyzed with robust regression analysis and a statistically significant regression model is created (F = 1.970; *p* = 0.047). When the BMI value increased by one unit, MBPS values increased by 1.62 units (*p* = 0.012). An increase of one unit in the FAR value was associated with a decrease of 0.264 units in the MBPS values. An increase of one unit in the FAR value was associated with a decrease in MBPS values. (*p* = 0.009). Other variables had no significant effect on MBPS (*p* > 0.050).

UPDRS motor scale median value 10.00; creatinine median value was 0.92 in PD; 0.70 in control group. Even both values were in normal limits the difference between the groups was statistically significant (*p* < 0.001) (PD 21.63 control 18.63). The difference of median values of MBPS between the groups was significant (*p* < 0.005). There was no significant difference between the groups according to age BMI, FAR, diastolic BP, night systolic BP, dipper and night diastolic BP, average BP (*p* > 0.050). There is also no difference between the groups concerning other parameters. (*p* > 0.050).

In PD group the effects of independent variables over morning surge are analyzed with Robust regression analysis and statistically significant regression model is created (F = 1.970; *p* = 0.047). When BMI value increased one-unit MBPS values increased 1.62 units (*p* = 0.012). When FAR value increases one-unit MBPS value decreases 0.264 units (*p* = 0.009). Other variables had no significant effect on MBPS (*p* > 0.050) (Figure 2).

We further examined the relations between the use of oral levodopa, dopamine agonists, and Jejuna dopa infusion with MBPS in the PD sample only. All Spearman correlation coefficients were non-significant (*p* > 0.050). The relation between Hoen–Yahr stage and hsCRP was also not significant. However, the association between Hoen–Yahr stage and MBPS was statistically significant (rho = 0.28, *p* = 0.048).

By ROC analysis, the area under the curve value was found to be 0.665, which is statistically moderately significant for differentiating Parkinson’s disease (*p* = 0.005). If the cut-off value is set at 25, Sensitivity = 48%, Specificity = 78%, PPV = 68.57%, and NPV = 60% (Table 3, Figure 3).

## 4. Discussion

The increasing prevalence of Parkinson’s disease (PD) is a significant concern, particularly as advancements in the treatment of PD and associated systemic diseases extend patient longevity. As the global population ages—an established risk factor for PD—the number of individuals requiring treatment and care is expected to rise, raising important social and economic implications [25,26]. Despite the long-standing recognition of PD, numerous questions remain unanswered. Research indicates that patients with PD are at a higher risk for myocardial infarction, congestive heart failure, stroke, and other cardiovascular causes of death compared to those without PD; however, the issue of blood pressure fluctuations in PD has received comparatively less attention [27,28]. Notably, high blood pressure has also been identified as a potential risk factor for developing PD [29]. While hypotension is typically anticipated in patients with autonomic disorders, studies have demonstrated that fluctuations in blood pressure and nighttime hypertension can occur in individuals with PD [30]. These fluctuations may stem from reduced baroreceptor sensitivity due to Lewy body infiltration in autonomic centers, including the dorsal motor nucleus of the glossopharyngeal and vagal nerves, as well as the age-related degeneration of these structures. Importantly, blood pressure abnormalities may manifest prior to the onset of motor symptoms [31]. The medications used to treat parkinsonism, and other cardiovascular conditions, can exacerbate blood pressure-related symptoms, potentially diminishing quality of life and leading to organ damage [32]. Relying on single daily blood pressure measurements provides limited information; thus, 24 h ambulatory blood pressure monitoring (ABPM) is crucial for capturing circadian blood pressure variations [33]. Tulba et al. reported in a meta-analysis involving 40 studies that abnormal blood pressure (938/2460), orthostatic hypotension in 38.68% (941/2433), supine hypertension in 27.76% (445/1603), and nocturnal hypertension in 38.91% (737/1894). Additionally, dipping status was often altered, with 40.46% of patients (477/1179) classified as reverse dippers and 35.67% (310/869) as reduced dippers [34].

In our study, only mean systolic blood pressure (MSBP) showed statistically significant differences between the patient and control groups. Due to the small size of the patient groups, comparisons regarding parameters such as Parkinson’s medications, comorbidities, and disease stage could not be made. Blood pressure changes in PD patients—particularly nighttime variations—are often overlooked. The complexity of pathophysiological processes and medication effects complicates the analysis of these changes. The mechanisms underlying nighttime hypertension and non-dipping patterns remain poorly understood. O’Brien et al. categorized dippers into normal dippers (≥10% to <20% fall) and extreme dippers (≥20% fall) [35]. Factors such as aging, sleep disorders, diabetes, heart failure, obesity, and chronic renal insufficiency may influence nighttime blood pressure regulation. High salt intake, increased natriuresis, and decreased parasympathetic reactivity during the night could contribute to abnormal blood pressure changes. Moreover, studies suggest that urine catecholamine excretion decreases at night in non-dippers compared to dippers. This phenomenon may lead to an increased sensitivity of α1 receptors and heightened vasoconstriction [36]. Other factors associated with non-dipping include short sleep duration, fragmented sleep patterns, and reduced sleep depth [37]. Both nighttime hypertension and non-dipping are considered predictors of cardiovascular events; however, some authors have reported low reproducibility of nocturnal dipping patterns. Consequently, before drawing definitive conclusions or planning therapeutic interventions related to blood pressure management in PD patients, more robust evidence must be gathered [38].

Our study did not reveal any significant differences in dipping patterns between the Parkinson’s disease (PD) and control groups. However, the median values for PD patients (4.96 mmHg; range: −20.75 to 22.59) were slightly lower than those of the control group (6.87 mmHg; range: −12.40 to 23.08) (Figure 2). Most of our patients were staged as Hoehn–Yahr I, suggesting that changes in dipping patterns may occur in later phases of the disease as sleep disorders and autonomic dysfunction become more pronounced. Additionally, the influence of medications on sleep and dipping patterns cannot be overlooked. Early morning blood pressure changes are closely linked to cardiovascular diseases, with myocardial infarctions, sudden deaths, and strokes frequently occurring in the morning hours due to neurohormonal changes that take place upon awakening [23]. The activation of the sympathetic nervous system and a decrease in vagal tone during this time contribute to increased blood pressure, heart rate, vasomotor tone, and blood viscosity. Conversely, MBPS values below 20 mmHg were likely not associated with increased cardiovascular risk; however, values exceeding 28 mmHg were identified as a potential risk factor for cardiovascular disease [27,39].

In our study, the median value of morning blood pressure surge (MBPS) in Parkinson’s disease (PD) patients (21.63 mmHg) was higher than that of the control group (18.63 mmHg). Analysis of independent variables indicated an inverse correlation between body mass index (BMI) and MBPS; specifically, for each one-unit increase in BMI, MBPS increased by 1.62 units. When setting the cutoff point for MBPS at 25 mmHg, we found sensitivity and specificity rates of 48% and 78%, respectively. The role of neuroinflammation in the pathophysiology of PD remains incompletely understood. However, dopamine deficiency may trigger inflammation by activating innate immune responses. The gut–brain axis is another proposed inflammatory pathway, suggesting that neuroinflammation affects both peripheral and central nervous systems. Elevated levels of serum and cerebrospinal fluid (CSF) cytokines such as IL-1β, IL-2, IL-6, IFN-γ, and TNF-α, along with increased counts of CD4+ lymphocytes, have been documented in PD patients [3]. Epidemiological studies examining the relationship between C-reactive protein (CRP) and Parkinson’s disease (PD) have yielded conflicting results. Some studies have identified a significant association between CRP levels and PD, while others have found no relationship when compared to control patients [40].

Furthermore, high variability in blood pressure can induce vascular inflammation. Inflammatory markers like fibrinogen, TNF-α, IL-6, and high-sensitivity C-reactive protein (hs-CRP) are associated with blood pressure changes [41]. The fibrinogen-to-albumin ratio (FAR) has been identified as a significant index in cardiovascular and cancer-related diseases [42,43]. Elevated FAR is recognized as an independent risk factor for exaggerated morning surges in patients with hypertension, suggesting that inflammation may play a role in the pathogenesis of MBPS [17]. However, our study did not find a positive association between FAR and MBPS (Figure 2). This lack of correlation may be attributed to the small sample size of both the patient and control groups, as well as the insignificance of FAR and CRP levels in our PD cohort. (In fact, the post-hoc power calculation for the present study was less than 0.80, indicating that the study was underpowered). There is a pressing need for further studies to examine additional peripheral inflammatory parameters in relation to blood pressure changes among PD patients. Given that blood pressure fluctuations are also related to autonomic functions, it would be beneficial to investigate the interplay between autonomic functions and inflammatory markers concurrently. A larger-scale, prospective study could provide clarity on these relationships. Notably, while creatinine levels showed significant differences that may indicate early signs of organ damage, they remained within normal limits (Table 2). Continuous monitoring will be essential to determine whether these variations impact prognosis concerning mortality and morbidity. In conclusion, our findings highlight the complexity of blood pressure dynamics in PD patients and underscore the need for comprehensive evaluations that incorporate both inflammatory markers and autonomic function assessments. As our understanding of these relationships evolves, targeted interventions may enhance patient outcomes.

According to the study by Hao-Min Cheng et al., an increased MBPS has been found to be associated with an increase in mortality. We wanted to investigate the increase in MBPS in Parkinson’s disease, which is the second most common neurodegenerative disease and has a rising prevalence in older age. PD is a disease that currently has multi-systemic effects associated with increased morbidity. We thought that MBPS could contribute to this increased morbidity risk [44].

Our study found that morning blood pressure surge (MBPS) levels were significantly higher in Parkinson’s disease (PD) patients compared to the control group, suggesting possible autonomic involvement (Figure 2). Given the complex pathophysiology of PD, characterized by neurodegeneration and autonomic dysfunction, it is crucial to investigate blood pressure changes thoroughly to prevent potential cardiovascular and cerebral events. To our knowledge, there are no similar studies in literature that specifically address this relationship. We recommend larger studies to further explore the connection between MBPS and clinical findings in PD patients.

One limitation of our study is the absence of echocardiographic evaluation, which could provide additional insights into cardiac function. Other limitations include the small sample size, the fact that some patients were already using antihypertensive medications, and the presence of diabetes mellitus (DM) in certain patient conditions known to increase the risk of autonomic dysfunction. Furthermore, we did not evaluate other non-motor complications of PD, such as orthostatic hypotension and supine hypertension, which could have influenced our findings.

## 5. Conclusions

In our study, only mean systolic blood pressure (MSBP) showed statistically significant differences between the patient and control groups. Our study did not reveal any significant differences in dipping patterns between the Parkinson’s disease (PD) and control groups.

Elevated FAR is recognized as an independent risk factor for exaggerated morning surges in patients with hypertension, suggesting that inflammation may play a role in the pathogenesis of MBPS [17]. However, our study did not find a positive association between FAR, hs-CRP, and MBPS. Our study found that morning blood pressure surge (MBPS) levels were significantly higher in Parkinson’s disease (PD) patients compared to the control group, suggesting possible autonomic involvement. We believe that there are several reasons for these results, as explained in the discussion section.

The association between Parkinson’s Disease (PD) and atherosclerotic disease emphasizes the detrimental effects of morning systolic blood pressure (MSBP) on cardiovascular and neurological health. Elevated MSBP is linked to increased end-organ damage in the heart and brain, potentially accelerating the progression of Parkinson’s Disease. This suggests that MSBP could represent a novel therapeutic target for preventing organ damage and reducing the risk of cardiovascular events. Future research should focus on conducting studies with larger patient populations while excluding confounding conditions that may influence blood pressure. Such investigations could clarify the role of MSBP in the context of both atherosclerotic disease and Parkinson’s Disease, providing insights into targeted therapeutic strategies. By addressing this morning surge in blood pressure, we may improve management approaches that consider both inflammatory processes and vascular health, ultimately enhancing patient outcomes in these interconnected conditions.

## Figures and Tables

**Figure 1 biomedicines-13-00363-f001:**
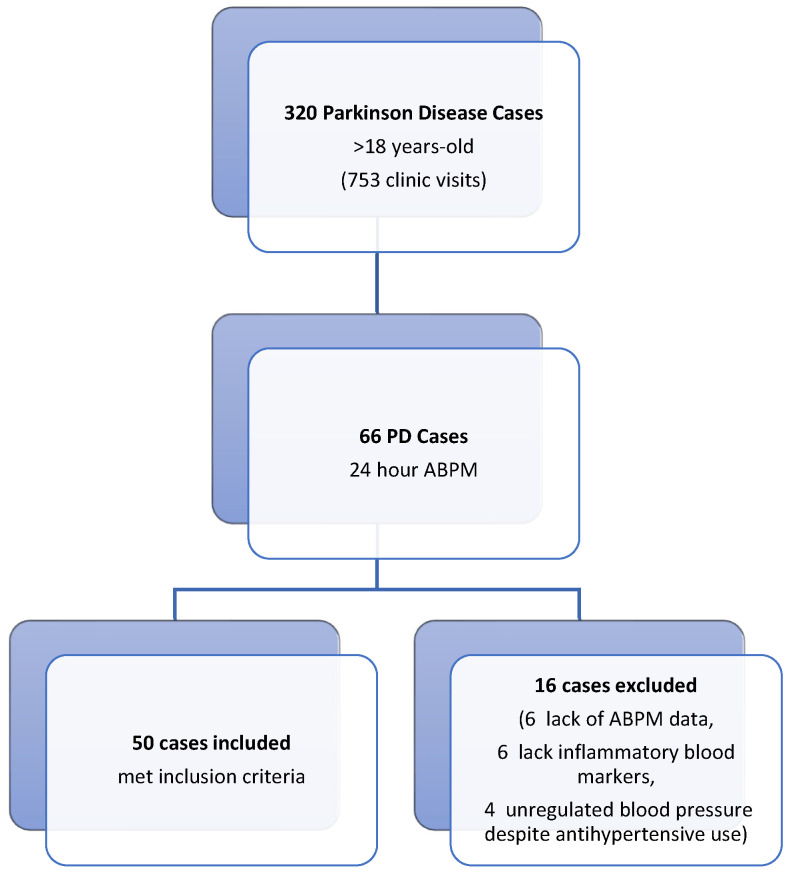
Flowchart of the patients included in the study.

**Figure 2 biomedicines-13-00363-f002:**
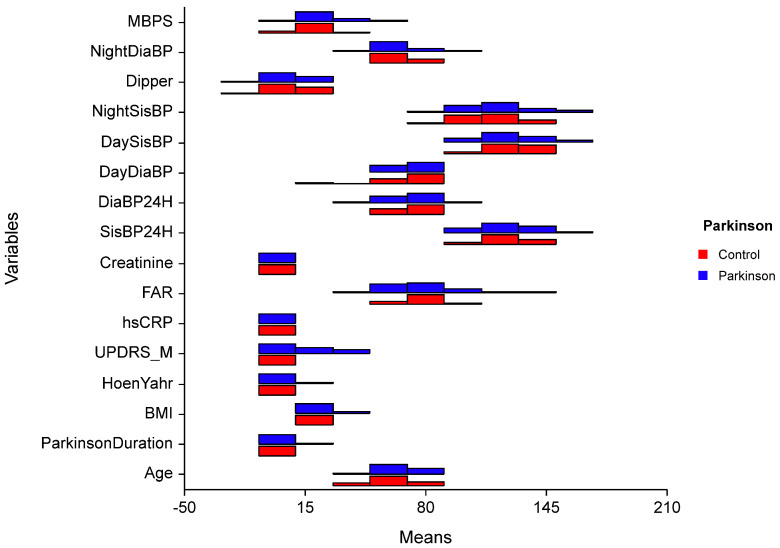
Comparison of quantitative data by groups. t: Independent two-sample t-test statistic; U: Mann–Whitney U, test statistic; UPDRS-M: Unified Parkinson’s Disease Rating Scale-Motor; BMI = Body Mass İndex; hsCRP = highly sensitive CRP; FAR = Fibrinogen Albumin Ratio; SisBP: Systolic Blood Pressure; DiaBP: Diastolic Blood Pressure; MBPS = Morning Blood Pressure Surge.

**Figure 3 biomedicines-13-00363-f003:**
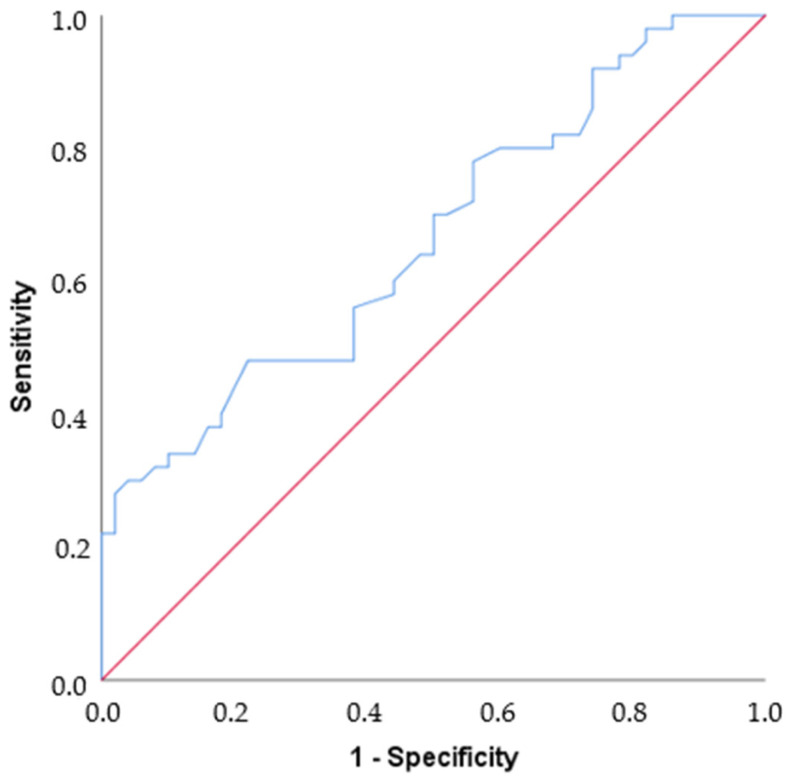
ROC curve. The blue line shows the ROC curve defined by the set of variables that predicts nearly 67% of the outcome (AUC = 0.665). The red line shows an example curve where only 50% of the outcome is correctly predicted by the study variables (AUC = 0.5).

**Table 1 biomedicines-13-00363-t001:** Distribution of categorical variables by groups.

	Controls	Patients	Total	Test Statistics	*p*
Sex					
Female	20 (40)	24 (48)	44 (44)	0.365	0.546 *
Male	30 (60)	26 (52)	56 (56)		
Accompanying illness					
No	50 (100)	39 (78)	89 (89)	---	---
Yes	0 (0)	11 (22)	11 (11)		
Oral L-dopa					
No	50 (100)	12 (24)	62 (62)	---	---
Yes	0 (0)	38 (76)	38 (38)		
Dopa agonist					
No	50 (100)	13 (26)	63 (63)	---	---
Yes	0 (0)	37 (74)	37 (37)		
Jejunal dopa					
No	50 (100)	46 (92)	96 (96)	---	---
Yes	0 (0)	4 (8)	4 (4)		
Hoen Yahr					
0	50 (100)	0 (0)	50 (50)	---	---
1	0 (0)	22 (44)	22 (22)		
1.5	0 (0)	3 (6)	3 (3)		
2	0 (0)	10 (20)	10 (10)		
2.5	0 (0)	2 (4)	2 (2)		
3	0 (0)	8 (16)	8 (8)		
4	0 (0)	5 (10)	5 (5)		

* Yates correction, frequency (percentage): Since all these parameters were evaluated in the patient group, comparison is not theoretically appropriate.

**Table 2 biomedicines-13-00363-t002:** The effects of independent variables over morning surge are analyzed with robust regression analysis.

Independent Variables	β1 (%95 CI)	S. Error	t	*p*
Fixed	24.659 (−32.292–81.61)	27.886	0.884	0.384
BMI	1.62 (0.389–2.85)	0.602	2.689	**0.012**
hsCRP	0.41 (−2.258–3.077)	1.306	0.314	0.756
FAR	−0.264 (−0.457–0.071)	0.095	−2.788	**0.009**
Creatinine	5.073 (−7.755–17.9)	6.281	0.808	0.426
Dipper	0.115 (−0.3–0.529)	0.203	0.565	0.577
Parkinson Duration	−0.632 (−1.75–0.487)	0.548	−1.153	0.258
SisBP 24 h	−0.063 (−0.325–0.199)	0.128	−0.491	0.627
DiaBP 24 h	−0.328 (−0.762–0.107)	0.213	−1.538	0.134
NightDiaBP	−0.025 (−0.397–0.346)	0.182	−0.139	0.890
Gender (Reference: Male)	5.667 (−2.217–13.551)	3.860	1.468	0.153
Comorbidity	3.89 (−2.877–10.656)	3.313	1.174	0.250
Dopa Oral	2.307 (−5.837–10.451)	3.988	0.578	0.567
Dopa Agonist	−1.81 (−10.717–7.097)	4.361	−0.415	0.681
Dopa Infusion	−11.291 (−24.839–2.257)	6.634	−1.702	0.099
Hoen Yahr				
1.5	−1.887 (−14.168–10.394)	6.013	−0.314	0.756
2	1.895 (−6.6–10.389)	4.159	0.456	0.652
2.5	−5.869 (−20.786–9.049)	7.304	−0.803	0.428
3	5.605 (−3.362–14.573)	4.391	1.277	0.212
4	11.3 (−1.045–23.646)	6.045	1.869	0.071

F = 1.970; *p* = 0.047, R^2^ = 0.555, β1: Unstandardized beta coefficient. BMI = Body Mass İndex. hsCRP = Highly sensitive CRP. FAR = Fibrinogen–Albumin Ratio. SisBP: Systolic Blood Pressure. DiaBP: Diastolic Blood Pressure. Bold means *p* < 0.050.

**Table 3 biomedicines-13-00363-t003:** Results of ROC analysis. PPV: Positive predictive value, NPV: Negative predictive value.

	AUC (%95 CI)	*p*	Cutoff	Sensitivity	Specificity	PPV	NPV
Morning Surge	0.665 (0.559–0.77)	0.005	25	48%	78%	68.57%	60%

## Data Availability

The data presented in this study are available on request from the corresponding author. The data are not publicly available due to privacy or ethical restrictions.

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
