# Peer review of "Relationship Between Morning Blood Pressure Surges and Peripheral Inflammatory Biomarkers in Parkinson’s Disease"

_biomedicines, 2025, doi:10.3390/biomedicines13020363_

Round 1

Reviewer 1 Report

Comments and Suggestions for Authors

In the research article entitled as ‘Relationship Between Morning Blood Pressure Surges and Pe-ripheral Inflammatory Biomarkers in Parkinson's Disease’, Ummu S. SARI et al tried to report an interesting relationship between morning blood pressure surges and peripheral Inflammatory Biomarkers (of Blood?) in PD. However, they didn’t describe these very well. It would be recommended to be published after revision. The concerns were provided as followed:

1.      The formation and description of title should be improved.

2.      What is the title of section 3.1? Please double check the integrality.

3.      The table seemed much difficult to understand the relationship. It would be better to use figure.

4. It would be satisfied with the figure to conclude the relationship and its significance.

Author Response

For review article

Response to Reviewer X Comments

1. Summary

Comments and Suggestions for Authors

In the research article entitled as ‘Relationship Between Morning Blood Pressure Surges and Pe-ripheral Inflammatory Biomarkers in Parkinson's Disease’, Ummu S. SARI et al tried to report an interesting relationship between morning blood pressure surges and peripheral Inflammatory Biomarkers (of Blood?) in PD. However, they didn’t describe these very well. It would be recommended to be published after revision. The concerns were provided as followed:

  1. The formation and description of the title should be improved.

Response 1: [We have changed font size error We cannot change the title because it needs to be the same as the title in the study plan, we submitted to the ethics committee.] Thank you for pointing this out. We agree with this comment.

  1. What is the title of section 3.1? Please double check the integrality.

Response 2: We've fixed the section numbering. (Line 173).

  1. The table seemed very difficult to understand the relationship. It would be better to use figures.

Response 3: We agree with this comment. We have added figure 2 instead of table 2. (Line 206)

  1. It would be satisfied with the figure to conclude the relationship and its significance.

Response 4: We agree with this comment. The necessary explanation is also provided in the results section of the text.

Reviewer 2 Report

Comments and Suggestions for Authors

This is a study of blood pressure (BP) fluctuations overnight and during the day in 50 control and 50 Parkinson's disease (PD) subjects. The details and results of BP monitoring and blood assays are presented in tabular form and analyzed for correlations and receiver operating characteristics (ROC). While the rationale for studying this problem (blood pressure surges upon awakening) and the data collection and analyses were acceptable, there are multiple notable deficiencies and limitations to this study:

1. The authors claim to have screened 753 PD subjects over one year (Aug 2021-Aug 2022) and selected 50 for inclusion in their study. How were these 50 selected? They are clearly a heterogenous group (30% had bilateral disease (H&Y scores 2.5-4.0)).

2. ~75% were taking L-DOPA (doses not specified), some were treated with jejunal L-DOPA, and ~75% also took dopamine agonist (which one(s)? doses (?)), nor were the dosing schedules standardized in relation to BP measurements. How many subjects took both L-DOPA and DA agonist?

3. PD patients vary as to "sleep benefit" (less PD symptoms upon awakening); was this factor part of their selection? 

4. The authors state that some were taking hypertensive medications. How could they include these subjects in their study, when they were taking medications that affect their primary variable?

5. The authors did not appear to address orthostatic hypotension (OH) quantitatively, a major "secondary" PD symptom. 

The major finding of the study, that PD subjects appear to have greater BP "surge" upon awakening is clinically important. Unfortunately, the reader is left wondering what this may have to do with PD as a multi-system disease in terms of when it appears in the disease course (ie., early unilateral vs. advanced bilateral disease), what its relation is (if any) to timing and amount of PD medications. The authors have invested a large amount of work in data collection and analysis, and I feel this paper can be salvaged with careful attention to the above. 

At several places in the manuscript the authors mentioned that their population was limited. I completely agree with this statement, and their study is more of a preliminary exploration of the problem of awakening blood pressure surges in PD. With their preliminary data, the authors can discuss the power limitations of their study and predict population sizes needed to demonstrate statistical significance.

Author Response

  1. The authors claim to have screened 753 PD subjects over one year (Aug 2021-Aug 2022) and selected 50 for inclusion in their study. How were these 50 selected? They are clearly a heterogenous group (30% had bilateral disease (H&Y scores 2.5-4.0)).

Reponse 1: Thank you for pointing this out. We agree with this comment. We have corrected the number of patients screened. For the patient group, we screened 320 Parkinson's disease (PD) (320 patients out of 753 patient visits) patients who visited the movement disorders outpatient clinic of the neurology department between August 2021 and August 2022 (Figure 1). Patients’ selection procedure is also explained in Figure 1. We could only include patients who already had 24 hours ABPM data. (Line 124 and 144)

  1. ~75% were taking L-DOPA (doses not specified), some were treated with jejunal L-DOPA, and ~75% also took dopamine agonist (which one(s)? doses (?)), nor were the dosing schedules standardized in relation to BP measurements. How many subjects took both L-DOPA and DA agonist?

Response 2: Thank you for pointing this out. We agree with this comment but No we couldn’t. Dopaminergic dosing schedules haven’t been standardized. Since the doses and timings of the dopaminergic treatment used by the patients individualized for each patient, so that a homogeneous patient group is not present. Therefore, the relationship between drug dosage and timing could not be evaluated in the study. As you pointed out it would be beneficial to investigate the effect of medications in prospective studies.

  1. PD patients vary as to "sleep benefit" (less PD symptoms upon awakening); was this factor part of their selection? 

Response 3:  Sleep benefit was not a part of patients’ selection criteria.

  1. The authors state that some were taking hypertensive medications. How could they include these subjects in their study, when they were taking medications that affect their primary variable?

Response 4: True but Only 3 patients had hypertension and all patients’ blood pressure were under control. (Figure 1) Patients with controlled blood pressure were included in the study. At the same time, since a significant number of Parkinson's patients also have hypertension, we thought it would not be appropriate to exclude these patients. The number of patients using antihypertensive medications is already low, and therefore, it is not statistically significant enough to impact the primary outcome.

  1. The authors did not appear to address orthostatic hypotension (OH) quantitatively, a major "secondary" PD symptom. 

The major finding of the study, that PD subjects appear to have greater BP "surge" upon awakening is clinically important. Unfortunately, the reader is left wondering what this may have to do with PD as a multi-system disease in terms of when it appears in the disease course (ie., early unilateral vs. advanced bilateral disease), what its relation is (if any) to timing and amount of PD medications. The authors have invested a large amount of work in data collection and analysis, and I feel this paper can be salvaged with careful attention to the above. 

At several places in the manuscript the authors mentioned that their population was limited. I completely agree with this statement, and their study is more of a preliminary exploration of the problem of awakening blood pressure surges in PD. With their preliminary data, the authors can discuss the power limitations of their study and predict population sizes needed to demonstrate statistical significance.

Response 5: As mentioned above, we included the patients who already had ABPM data because of that we could not provide quantitative information about orthostatic hypotension.

Indeed, the post-hoc power analysis showed that the number of patients included in our study is not sufficient to have the power to reach a certain conclusion. Additional work to increase the number of patients will be beneficial to overcome this limitation.

We have made additions to the discussion section related to this question. (Line 337-342)

In addition to the conclusion section, we mentioned possible health impairments due to Increased MS and discussion. However, we expanded the information about the increased morning surge based on the previous reports.

Reviewer 3 Report

Comments and Suggestions for Authors

The manuscript submitted to Biomedicines by Gulseren BUYUKSERBETCI and co-authors is devoted to analyzing the relationship between morning blood pressure surge and inflammatory markers in patients with Parkinson's disease.

The authors' choice of fibrinogen-to-albumin ratio and high-sensitivity C-reactive protein (hs-CRP) as inflammatory markers requires detailed justification.

The reviewer has some comments that should be addressed before the manuscript can be recommended for publication

1) The introduction contains a single paragraph - lines 35-101. It should be divided into several parts.

2) The choice of fibrinogen-to-albumin ratio and high-sensitivity C-reactive protein as markers of inflammation should be justified in the Introduction.

3) The Introduction should state the hypothesis that the authors tested in their study, and the Conclusion should answer the question of whether the hypothesis was proven or disproven.

4) The article contains no illustrative material, except for the graph in Figure 1, which can hardly be called informative. From the material presented in the Tables, it is necessary to select what can be presented in diagrams or graphs and replace these data in graphical form.

5) Table 2 should be changed, perhaps some of the material should be moved, for example, min-max to the Supplementary. In its current form, it is impossible to get any information from it. The same is relevant for Table 3.

6) The Discussion section should clearly and thoroughly explain the physiological significance of the observed correlation between morning blood pressure and inflammatory markers, with references to the literature.

I believe that the article needs major revisions before it can be considered for publication in Biomedicines.

Author Response

Comments 1) The introduction contains a single paragraph - lines 35-101. It should be divided into several parts.

Response 1: Thank you for pointing this out. We agree with this comment. Therefore, we have divided the introduction part into parts.

Comments 2) The choice of fibrinogen-to-albumin ratio and high-sensitivity C-reactive protein as markers of inflammation should be justified in the Introduction.

Response 2: we agree with this comment. We have made additions to the introduction section related to this question. (Line 101-112)

Comments 3) The Introduction should state the hypothesis that the authors tested in their study, and the Conclusion should answer the question of whether the hypothesis was proven or disproven.

Response 3: We agree with this comment. We have made additions to the introduction and conclusion section related to this question.

Comments 4) The article contains no illustrative material, except for the graph in Figure 1, which can hardly be called informative. From the material presented in the Tables, it is necessary to select what can be presented in diagrams or graphs and replace these data in graphical form.

Response 4: True. We have changed complicated Tables and added Figure 2.

Comments 5) Table 2 should be changed, perhaps some of the material should be moved, for example, min-max to the Supplementary. In its current form, it is impossible to get any information from it. The same is relevant for Table 3.

Response 5: True. We have changed complicated Tables and added Figure 2.

Comments 6) The Discussion section should clearly and thoroughly explain the physiological significance of the observed correlation between morning blood pressure and inflammatory markers, with references to the literature.

Response 6: Thank you for pointing this out. We agree with this comment. Therefore, we have made additions to the discussion section related to this question.

Round 2

Reviewer 1 Report

Comments and Suggestions for Authors

I would recommend it to be published.

Author Response

Thank you very much for your diligent efforts and recommendation for publishing.

Reviewer 2 Report

Comments and Suggestions for Authors

This is a critique of a revised version of a manuscript of which I reviewed the original version. The authors have not adequately addressed my original concerns, and their responses are inadequate for a clinical research paper that required a more standardized subject selection, medication details standardization (particularly since the medications used for PD can affect BP), and exclusion of ANY subjects using anti-hypertensive medications. For these reasons, I find this paper unacceptable as written.

This is unfortunate, since the authors are attempting to address a clinically important topic- BP surges upon awakening in PD. They need a much more sophisticated characterization of their subject population (not just inclusion based on whether or not they had an ambulatory BP monitor (ABPM) data set). They selected 50 PD subjects out of >300.

They do provide hints that PD is associated with increased BP upon awakening, but the heterogeneity and lack of standardization of their subject population makes it impossible to draw any conclusions about PD and morning BP surges. That they can demonstrate statistical significance of some findings is not enough when the PD subject population is so heterogenous.

Some questions to consider:

1. Did all subjects experience increased morning BP, or was there any relationship to L-DOPA only, agonist only, or L-DOPA + agonist? H&Y stage?

2. Was there any relationship among PD stage (H&Y number) and CRP?

3. Almost half (44%) of their PD population had very early disease (H&Y = 1.0). This group should be analyzed separately, and those with bilateral disease (H&Y 2.5-4.0) could form a separate group.

Author Response

We appreciate your constructive suggestions and advice. We tried to make some improvement according to suggestions. Unfortunately, we were not able to fully implement all your suggestions.

We considered questions 1 and 2 and performed a bivariate correlation analysis. However, we decided not to pursue the third question because it would reduce the power of the analysis.

We further examined the relations between the use of oral levodopa, dopamine agonists and Jejuna dopa infusion with MBPS in the PD sample only. All Spearman correlation coefficients were nonsignificant (p>0,050). The relation between Hoen Yahr stage and hsCRP was also not significant. However, the association between Hoen Yahr stage and MBPS was statistically significant (rho=0.28, p=0.048). (lines between 240-243).

Reviewer 3 Report

Comments and Suggestions for Authors

I believe it would be useful for a potential reader of the article if the authors added a formulation of the hypothesis to the Introduction section and whether it was true to the Conclusion section.

Author Response

Thank you for your constructive suggestions and advice. We tried to make some improvement according to suggestions.  (Lines 110-119 and 380-390).

Round 3

Reviewer 2 Report

Comments and Suggestions for Authors

This is the third revision of a paper I have previously reviewed. The authors have responded to my previous concerns and have admitted that their study has limitations in terms of PD subject selection and limited number of PD subjects. They admit that they did not control for PD medications and selected PD subjects on the basis of whether they had undergone 24 hr continuos monitoring of BP. Thus, their study is not prospective in terms of testing a hypothesis. As a result, they can only report statistically significant conclusions to a very limited variable set. They admit that their study was under-powered to assay significance of several variables. At best, theirs is a limited study which will not change much in the clinical world of managing PD.

While their study has many limitations, many of which they have discussed in their text in this third revision, they are addressing a clinically important problem (morning BP surges (MBPS) in PD). Hopefully, this paper will stimulate them (and/or others) to pursue the question of MBPS in PD in a more controlled, prospective manner.

English is fine, and paper reads well.